# Dietary Intake, Serum Hormone Concentrations, Amenorrhea and Bone Mineral Density of Physique Athletes and Active Gym Enthusiasts

**DOI:** 10.3390/nu15020382

**Published:** 2023-01-12

**Authors:** Jaakko Mursu, Maija Ristimäki, Inga Malinen, Pirita Petäjä, Ville Isola, Juha P. Ahtiainen, Juha J. Hulmi

**Affiliations:** 1Faculty of Sport and Health Sciences, NeuroMuscular Research Center, University of Jyväskylä, 40014 Jyväskylä, Finland; 2Institute of Public Health and Clinical Nutrition, University of Eastern Finland, 70211 Kuopio, Finland; 3Department of Food and Nutrition, University of Helsinki, 00100 Helsinki, Finland

**Keywords:** fitness, low energy availability, relative energy deficiency in sport (RED-S), menstrual status, sport

## Abstract

As the diet, hormones, amenorrhea, and bone mineral density (BMD) of physique athletes (PA) and gym enthusiasts (GE) are little-explored, we studied those in 69 females (50 PA, 19 GE) and 20 males (11 PA, 9 GE). Energy availability (EA, kcal·kgFFM^−1^·d^−1^ in DXA) in female and male PA was ~41.3 and ~37.2, and in GE ~39.4 and ~35.3, respectively. Low EA (LEA) was found in 10% and 26% of female PA and GE, respectively, and in 11% of male GE. In PA, daily protein intake (g/kg body mass) was ~2.9–3.0, whereas carbohydrate and fat intakes were ~3.6–4.3 and ~0.8–1.0, respectively. PA had higher protein and carbohydrate and lower fat intakes than GE (*p* < 0.05). Estradiol, testosterone, IGF-1, insulin, leptin, TSH, T4, T3, cortisol, or BMD did not differ between PA and GE. Serum IGF-1 and leptin were explained 6% and 7%, respectively, by EA. In non-users of hormonal contraceptives, amenorrhea was found only in PA (27%) and was associated with lower fat percentage, but not EA, BMD, or hormones. In conclusion, off-season dietary intakes, hormone levels, and BMD meet the recommendations in most of the PA and GE. Maintaining too-low body fat during the off-season may predispose to menstrual disturbances.

## 1. Introduction

Physique sports include various divisions from bikini athletes to bodybuilders, in all of which competitors strive to achieve an aesthetic appearance with symmetry, balance and muscle “definition” achieved with minimal fat mass. Competitive physique athletes have an off-season and a competition preparation phase. During a typical 3–8-month contest, preparation athletes aim for low body fat with negative energy balance while maintaining muscle size with resistance training and high protein intake [1]. In competitions, physique athletes are judged by their aesthetic appearance, and a relatively high amount of muscle mass and low levels of body fat are preferred [1].

Physique athletes spend most of their time in the off-season (also called ‘improvement season’), where the main goal is to increase muscle size while minimizing body fat accumulation [2]. This is achieved with resistance training providing mechanical stimuli [3] and positive energy balance, adequate protein and carbohydrate intake which support intensive exercise, muscle growth and recovery [4,5,6]. For experienced trainees, a 5–10% energy surplus, i.e., 200–300 kcal above daily maintenance, is recommended which is estimated to increase body size by 0.25% per week [2] and to minimize the adverse health effects of dieting, e.g., on hormonal functions. Not all, however, follow the nutrition recommendation in the off-season, and these strategies are common without scientific evidence, which can expose physique athletes to unwanted health or performance-related consequences [7]. During the off-season, obtaining optimal energy intake may be psychologically challenging, especially after a long competition preparation. In female physique athletes, body dissatisfaction and fear of gaining fat may be more common compared to men [8].

Low energy intake may negatively affect gains in muscle size, although strength gains seem to be less affected [9]. Low energy intake may also affect the health of athletes. The adequacy of energy intake can be assessed by calculating energy availability (EA) which estimates the amount of energy available, e.g., for body functions and training adaptation. EA is derived by subtracting exercise energy expenditure from energy intake and dividing by fat-free mass (FFM). EA ≥ 40–45 kcal/kg of FFM is considered to be optimal during the improvement season, supporting performance, training adaptation and health [10]. In contrary, low EA (<30 kcal/kg of FFM) is necessary for weight loss, but may negatively affect the performance, the production of various hormones and bone health [10,11] and thus may lead to a multisyndrome condition called relative energy deficiency in sport (RED-S) [10]. Studies investigating weight loss in these athletes have shown that low EA may suppress leptin, triiodothyronine (T3), testosterone, and estradiol concentrations, and increase the incidence of menstrual irregularities, including amenorrhea [12] markers of immunosuppression [13], and adaptive thermogenesis [14]. Furthermore, amenorrhea, which is considered to be a long-term marker for low EA [10], impairs bone health, increases the risk for bone stress injury and cardiovascular disease [11], and may thus, through these effects, alter athletes’ performance and health.

These changes are reversible if the energy intake is increased to optimal levels [10]. However, the studies on these athletes have focused on competition preparation diets and not the off-season phase. Thus, there is not much evidence about the dietary habits and physiology of the athletes during this phase of their training season and how they compare to active gym enthusiasts who often have similar goals, except they may not have the aim to compete.

Currently, very little is known about the dietary habits and related health of the physique athletes, especially in the off-season, and how they compare to those who actively train in the gym but do not compete in any physique or strength sports. Therefore, the aim of our cross-sectional study was to assess dietary intake, serum hormone concentrations, amenorrhea, and bone mineral density of Finnish physique athletes in the off-season and in gym enthusiasts who have similar training goals but have no experience in competing or competition preparation in physique sports. Additionally, our aim was to assess whether amenorrhea is associated with EA, body composition, training background, serum hormone concentrations, and bone mineral density.

## 2. Materials and Methods

### 2.1. Overall Approach to the Problem

The current study is a sub-study of a larger Physique Athlete Study conducted by the University of Jyväskylä including a cohort collected in 2015–2016 [12,13,14,15] and 2019–2020 (Isola et al. submitted for revision, ClinicalTrials.gov ID: NCT04392752). The current study looked at the prevalence of energy availability and the distribution of intake of different energy nutrients. Energy availability was compared with measurements of hormone levels and bone density taken at the same time point and menstrual status in a cross-sectional setting.

### 2.2. Participants

A total of 184 healthy, physically active young females in study 1 and 49 males and 40 females in study 2, recruited by web page and social media advertisements, volunteered to participate in the study via the university web page, the governing sports body web page for physique sport, and associated social media. They were physique athletes or active gym enthusiasts volunteered to participant in either a weight loss and weight gain regimen or weight maintenance.

The participants who were diagnosed with chronic diseases; reported using prescribed medications excluding birth control pills or any substances or methods prohibited by the World Antidoping Agency (WADA) such as performance-enhancing drugs; competing in junior (below 19 years of age) or master (over 40 years of age) categories; or competing at a non-drug-tested competition were excluded from the study. Participants who had competed within six months before the measurements were also excluded from the study. An online pre-study questionnaire was sent to the volunteers who claimed to meet the study’s inclusion criteria. Additional inclusion criteria for the physique athlete group were competition background in physique sports before the start of the study or registration for competitions in the following months under the International Fitness and Bodybuilding Federation (IFBB). Moreover, the inclusion criteria for the included divisions of IFBB athletes were the following: classic bodybuilding and men’s physique in males and fitness, body fitness, bikini fitness, wellness fitness and women’s physique in females. The gym enthusiasts group consisted of participants who also volunteered for the physique sport study and were actively and purposefully training to improve their physique but had not previously participated in competitions or did not intend to participate competitions in the near future (see more details of physique athletes and gym enthusiasts in Table 1). All the gym enthusiasts had a background of at least two resistance training years and they currently all train in a goal-oriented manner (improving their physique) about three to four times in the gym per week. The participants selected for the study completed an additional questionnaire that was subsequently reviewed by the study physician. The final study involved, in total, 89 participants; 61 were physique athletes and 28 gym enthusiasts. Out of 69 female participants, 50 were physique athletes and 19 were gym enthusiasts, and out of 20 male participants, 11 were physique athletes and 9 were gym enthusiasts (Table 1).

All participants gave their written informed consent for inclusion before they participated in the study (first cohort) and the Ethics Committee of the Central Finland Health Care District (19U/2018), Finland (the second cohort). The latter cohort was also registered at ClinicalTrials.gov ID: NCT04392752. The studies were conducted according to the guidelines of the Declaration of Helsinki. The participants were given comprehensive explanations regarding the study design, protocols, and possible risks. All participants gave written informed consent. The participants were given identification numbers, and the research group was blinded throughout the study.

### 2.3. Study Design

The study was a cross-sectional analysis of an observational study. We provided no intervention, and all groups followed their own preferred diet and exercise regimen. The time point for the measurement was during the off-season before a possible competition season dieting phase. The participants arrived at the laboratory between 07.00 and 09.00 am after at least eight hours of fasting and after instruction to sleep at least eight hours and abstain from alcohol and caffeine for 12 h, and exercise for 24 h. If participants traveled over 50 km to the laboratory, they were provided a hotel room for the night before the measurement day. Participants were advised to avoid physical activity, such as walking, jogging, and cycling, on the morning of assessment. The participants from the hotel were transported to the laboratory by car. In comparison, participants who lived closer than 50 km to our laboratory were advised to come via car or public transportation to the tests.

### 2.4. Estimation of Energy Availability

EA was calculated as (energy intake—exercise energy expenditure)/FFM. In the study, body composition was measured by DXA, InBody720 bioimpedance and skinfold measurement, of which DXA was chosen as the body composition method for EA (more below).

Study participants kept a 4-day food diary or provided information of their current dietary program by their coaches that they strictly followed. These included dietary supplements and all beverages, except water. The intake of protein and other energy providing supplements was included in the calculations of dietary energy and nutrients. The food diaries and the completed nutrition programs were analyzed by nutrient analysis software (Aivodiet, Flow-team Oy, Oulu, Finland) that is based on the national Food Composition Database Fineli in Finland maintained by the Finnish Institute for Health and Welfare. The software uses Fineli, which is a database maintained by the Finnish Institute for Health and Welfare (THL).

Exercise energy expenditure was estimated using exercise diaries and exercise programs. Subjects exercised according to their own exercise programs throughout the study period. They were asked to report all exercise in the exercise diary provided to them. The exercise diaries were used to determine (1) the number of resistance exercise workouts per week and type of workouts (upper body/lower body/whole body) and (2) the number of aerobic workouts per week, type and duration. For some subjects, exercise data were collected directly from their exercise programs.

For each exercise, a metabolic equivalent (MET) value was determined according to Ainsworth et al. [16,17] The MET coefficient indicates the multiple of the energy expenditure compared to the resting energy expenditure. In this study, the MET values for resistance exercise were used as follows: lower-body exercise 6, upper-body exercise 4 and total-body exercise 5 MET. Resistance exercise duration was set at 1 h for those who did not report exercise duration separately, referring to previous studies in physique athletes/bodybuilders [18].

Exercise-induced energy expenditure was calculated by multiplying the MET value by the duration of the exercise to obtain METh values for each exercise. The METh values for each exercise in a week were summed and divided by seven to give the average METh value for a single day’s exercise. This number was multiplied by the body weight of each participant to obtain the energy expenditure in kilocalories (kcal) spent due to exercise.

### 2.5. Body Composition and Bone Parameters

Body composition was estimated by dual-energy X-ray absorptiometry (DXA, Lunar Prodigy Advance EnCore version 14.10.022, GE Medical Systems—Lunar, Madison, WI, USA) after overnight fasting. The participants were measured with their arms at their sides with minimal clothing (i.e., underwear). Their legs were secured by non-elastic straps at the ankles. All metal objects were removed from the participant before the scan. DXA measurements were used to record participants’ weight, fat percentage, fat-free mass (FFM), bone mineral density (BMD) and z-score. The z-scores compare BMD of individuals to age and sex-matched controls and was thus used in the present study as a measure of BMD. A z-score below −2.0 is low bone density, being, thus, below the expected range for age and sex, but in athletes, even −1.0 has been considered to warrant further investigation [19].

### 2.6. Serum Hormones and Menstrual Status

Venous blood samples were obtained from the antecubital vein into serum tubes (Venosafe; Terumo Medical Co., Leuven, Hanau, Belgium) using standard laboratory procedures. Samples were stored in room temperature for 30 min, after being centrifuged at 3500× *rpm* for 10 min (Megadure 1.0 R Heraeus; DJB Lab Care, Hanau, Germany). Free thyroxine (T4), free triiodothyronine (T3), thyroid-stimulating hormone (TSH) insulin and estradiol were analyzed from serum with the Immunolite 2000 XPi, immunoassay system (Seimen Healtineers, Erlangen Germany) using Immulite^®^ 2000 Free T3 (L2KF32), Immulite^®^ 2000 Free T4 (L2KFT42), and Immulite^®^ 2000 Third Generation TSH (L2KTS2), insulin Immulite^®^ 2000 (L2KIN-19), and estradiol Immulite^®^ 2000 (L2KE2-17) commercial kits. Serum leptin was analyzed with the Dynex Ds 2 ELISA processing System (DYNEX Technologies, Chantilly, VA, USA) using a commercial kit (Human Leptin ELISA, Clinical Range, REF RD191001100. These hormones are routinely analyzed in our laboratory and day-to-day reliability (CV%) for all of these hormones in our laboratory is <8%.

Menstrual status and the possible use of hormonal contraception (oral hormonal contraceptives or intrauterine device) was investigated using questionnaires. The participants returned the questionnaire at the time of measurement.

### 2.7. Statistical Analysis

The females and males were divided into three categories according to EA: (1) <30 kcal/kg FFM, (2) 30–45 kcal/kg FFM and (3) >45 kcal/kg FFM and (1) < 30 kcal/kg FFM, (2) 30–40 kcal/kg FFM and (3) >40 kcal/kg FFM, respectively [11]. Female participants were divided into three categories according to menstrual status: (1) no menstrual period for more than 6 months; (2) at least one menstrual period in the last 6 months; (3) no menstrual period for more than 6 months but using hormonal contraception and (4) at least one menstrual period in the last 6 months and using hormonal contraception.

The study data were analyzed using IBM SPSS statistical software 28.0 (Armonk, New York, USA) and the ones visualized also with GraphPad Prism version 9.4.1 (GraphPad Software, San Diego, California, USA), with a *p*-value < 0.05 as the limit of statistical significance. Descriptive analyses were used to examine baseline data, health variables, energy nutrients, energy intake and exercise data. Results were described using means and standard deviations. The effects of sex and group on age, anthropometrics, training volumes, exercise duration, MET hours, exercise energy expenditure, dietary intakes, EA, hormones and BMD were examined using two-way ANOVA. Percentages were used to describe categorical variables and chi-square tests were used in statistical comparison. Differences between groups were examined using a two-tailed *t*-test for normally distributed variables and a Mann–Whitney U test, the unpaired counterpart of a two-tailed *t*-test for non-normally distributed variables. The Kolmogorov–Smirnov and Shapiro–Wilk tests were used to test the normality of the variables. The association of energy availability with bone density and hormone levels was examined using Pearson’s correlation coefficient and variables where a correlation was found were tested using linear regression analysis. 

## 3. Results

BMI and fat-free mass (FFM) were higher and body fat percentage lower in males than in females (Table 1). Within sexes, physique athletes and gym enthusiasts were similar in age, height, weight, BMI, body fat and FFM, although higher FFM in physique athletes reached borderline significance when compared to gym enthusiasts (*p* = 0.054, Table 1). The number of training sessions per week, total amount of physique training per week, and total METs were higher in females, but the number of training years was shorter than in male physique athletes (*p* < 0.05, Table 1). Resistance training volume and total training volume tended to be higher in physique athletes compared to gym enthusiasts (Table 1), mainly due to the higher volumes in female physique athletes when compared gym enthusiasts (*p* < 0.05). No difference was found in the number of training sessions, training years, MET, or exercise energy expenditure between athletes and gym enthusiasts (Table 1).

The EA was higher in females than in males (*p* = 0.020), while no difference was found between athletes and gym enthusiasts in EA (*p* = 0.392) (Figure 1A). The EA was in the optimal range (in females >45 kcal/kg FFM and in males >40 kcal/kg FFM) in 30% of the female and in 18% of the male physique athletes, and in 32% of the female and in 11% male gym enthusiasts, respectively. The prevalence of low energy availability (LEA) in female physique athletes was 10%, while in male physique athletes, no LEA was found (*p* = 0.274). In female and male gym enthusiasts, the prevalence of LEA was 26% and 11%, respectively (*p* = 0.360). Relative to bodyweight, physique athletes had higher dietary intakes of protein and carbohydrates, and a lower intake of fat (*p* < 0.05) than gym enthusiasts (Figure 1 B–D). Female athletes had a lower intake of carbohydrates per body weight (*p* = 0.015), but higher dietary fat intake per body weight (*p* = 0.013) compared to male athletes, while in gym enthusiasts, carbohydrate intake did not differ between sexes. Within sexes, female physique athletes had higher intake of protein per body weight (*p* = 0.005) and lower intake of fat per body weight (*p* = 0.002) than female gym enthusiasts. The findings were similar, although not statistically significantly different, between male athletes and gym enthusiasts.

Due to the uncertainty whether dietary and exercise parameters are able to estimate EA, we next analyzed blood hormones [20]. No difference was found in estradiol, testosterone, IGF-1, insulin, leptin, TSH, T4, T3 or cortisol concentrations between physique athletes and gym enthusiasts within sexes (Figure 2). In athletes and gym enthusiasts, the hormone profiles were similar and in most, the concentrations were within reference values. The exception was cortisol in females, which was higher than reference range in 48% of the females. When comparing female athletes (46%) and gym enthusiasts (53%), the proportion of the females with higher cortisol concentrations than reference range were similar. In other hormones, the proportions out of reference were smaller and similar between athletes and gym enthusiasts.

Long-term LEA can lead to low bone mineral density (BMD) [20]. Bone mineral density was similar in female and male physique athletes when compared to gym enthusiasts (Figure 3A), while in males, the BMD (1.33 ± 0.08 g/cm^2^) was higher than in females (1.23 ± 0.08 g/cm^2^, *p* < 0.001). The z-scores were on average very good and only one physique athlete man had a z-score <−1.0 warranting further investigation (Figure 3B).

Female athletes who have low BMD also often report menstrual dysfunction [20]. In all athletes, 28.6% reported to have no menstrual bleeding, while the proportion was 11.1% in gym enthusiasts, respectively. As reported amenorrhea is unreliable in those who use hormonal contraceptives, we chose to analyze amenorrhea only in those who did not use hormonal contraceptives. The prevalence of amenorrhea in those female physique athletes was 27% (6/22), while in the gym enthusiasts, no amenorrhea was reported (*p* = 0.099). When comparing female physique athletes who did not use hormonal contraceptives and had amenorrhea to those with menstruation, there were no differences in BMD, EA, and hormonal concentrations (Figure 4). However, athletes with amenorrhea had lower fat percentage and more resistance training years (Figure 4). Additionally, serum testosterone tended to be (*p* = 0.10) lower in amenorrheic females (Figure 4M).

Finally, we assessed possible associations between the measured variables. Out of the measured hormones, EA correlated significantly with IGF-1 (*r* = 0.272, *p* = 0.010) and leptin (*r* = 0.281, *p* = 0.008) concentrations. However, in regression analysis, EA explained only a small proportion of the variation in IGF-1 (6%) and leptin (7%) concentrations, and the former association was found only in females. Out of macronutrients, protein intake per body weight correlated with IGF-1 (*r* = 0.225, *p* = 0.034) and explained 4%.

## 4. Discussion

In our study, we assessed the dietary intakes, serum hormonal concentrations, amenorrhea, and BMD of female and male physiques competitors in the off-season. Currently, very little information is available from this period, while most of the studies have focused on the competition preparation period and potentially overexaggerated low EA interpretations in physique sports. In the off-season, physique athletes would ideally have dietary intake supporting training, protein synthesis, and health, including normal hormonal functions [1,2]. Furthermore, we compared physique competitors to non-competitors who were goal-oriented gym enthusiasts without a competition background or plan to compete in physique sports in the near future. The age, weight, body composition, and training were very similar in competitors and gym enthusiasts within the sexes, and the only significant difference was that the female competitors had a higher training volume than non-competitors.

In our experience, the cohort of physique athletes and gym enthusiasts typically report their dietary intakes exceptionally carefully [21]. We found that the dietary intake of the physique competitors, and also of the gym enthusiasts, was close to those recommended for physique athletes [2]. Energy availability ≥45 kcal/kg of FFM is considered to be optimal for female athletes and ≥40 kcal/kg of FFM for male athletes [11]. Recommended high EA supports training, protein synthesis, recovery, and health, while the effects of LEA are contrary, predisposing to adverse effects of exercise performance, protein synthesis, recovery, and health [11]. In our study, the EA was on average ~41 kcal/kg FFM/d in female physique athletes and ~37 kcal/kg FFM/d in male athletes. The recommended 45 kcal/kg of FFM for female and 40 kcal/kg of FFM for male athletes would have required an additional ~200 kcal, which is a relatively low amount of energy, an equivalent of, e.g., a small snack. The energy availability was slightly higher in physique athletes compared to gym enthusiasts (39 kcal/kg FFM in females and 35 kcal/kg FFM in males), although the difference was not statistically significant. Additionally, the prevalence of LEA in female physique athletes was low (10%) while in male athletes, no LEA was found. The LEA tended to be more prevalent in gym enthusiasts. This supports the notion that physique athletes emphasize sufficient energy intake equal to or perhaps more than non-competitors. Together, these results indicate that the energy availability of physique athletes appears to be adequate during the off-season and close to recommendations and at least as good as in gym enthusiasts without a background or future competitive goals in physique sports.

To our knowledge, the BMD of physique athletes has received very little scientific attention. In our study, BMD was similar in female and male physique athletes compared to gym enthusiasts and was higher in males than females, as expected. Additionally, the z-scores were on average very good. We also found that most of the hormone levels in all individuals were within reference values, and no difference was found in estradiol, testosterone, IGF-1, insulin, leptin, TSH, T4, T3, or cortisol concentrations between physique athletes and gym enthusiasts within sexes. In 10–16% of female athletes, the concentrations of the mentioned hormones were below, while almost 50% had cortisol concentrations above the reference range which, although being higher, agrees with the previous studies in athletes who often have high serum cortisol [22]. This may be in part due to early sample collection in this study (often at 7–8 am) as serum cortisol is highest in the morning 20–45 min after waking up [23]. That is a time-point when many individuals had blood samples taken, which can be speculated to be slightly earlier on average than in laboratory references. In general, when compared to non-competing females, the proportions of hormones below and above the reference range were almost identical. This suggests that competing in physique sports does not have a long-term effect on hormone concentrations, and most likely, these hormone functions recover during the off-season. However, because of cross-sectional analysis and the observational nature of the study, we cannot verify this. We have previously shown that restrictive and long-term dieting may suppress leptin, triiodothyronine (T3), testosterone, and estradiol concentrations and increase the incidence of menstrual irregularities, including amenorrhea [12]. However, these changes are reversible if the energy intake is increased to pre-competitive diet levels [12]. Body weight and the concentrations of all hormones except T3 and testosterone of physique competitors returned to baseline during a 3–4-month recovery period. Further studies with larger sample sizes are warranted to investigate the possible physiological effects of repeated, e.g., annual, preparation to physique competitions in female and male athletes.

A total of 27% of the female athletes who did not use hormonal contraceptives had amenorrhea, which may be a result of long-term LEA [10]. However, these females were not the same individuals with LEA, according to dietary information and supported also by other studies [24]. In female gym enthusiasts, no amenorrhea was reported. When comparing amenorrheic individuals to non-amenorrheic, we found no differences in BMD, EA, or hormonal concentrations. However, athletes with amenorrhea had a lower fat percentage and longer resistance training history (Figure 4). Additionally, serum testosterone tended to be (*p* = 0.10) lower in amenorrheic females. These results suggest that amenorrhea in competitors may be a result of maintaining too-low body fat [12], as amenorrheic athletes have earlier been reported to have elevated drive for thinness compared to eumenorrheic athletes [10], although we have no further data to support this hypothesis. On the other hand, the amenorrhea may also be due to a previous disruption of the menstrual cycle caused by preparations for the competition.

In previous studies in physique (fitness) athletes (*n* = 25), the prevalence of amenorrhea was 8% before the competition preparation period and was increased to 24% two weeks before the competition and remained the same 1 month after the competition [25]. Similarly, we observed in an earlier study that during the competition preparation, menstrual irregularities and amenorrhea increased [12]. In addition, Halliday et al. [26] reported in their case study that menses was absent for 71 weeks in a female physique athlete after the competition. Long periods of LEA predispose to amenorrhea, and despite current EA and hormonal concentrations being mainly normal, the recovery of testosterone concentration and menses may require a longer time. When comparing with other athletes in previous studies, the prevalence of amenorrhea found in PA in our study was similar. In a recent review, the prevalence of primary amenorrhea in rhythmic gymnastics was 25%, in soccer 20%, and in swimming 19%; and secondary amenorrhea in cycling was 56%, in triathlon 40%, and in rhythmic gymnastics 31% [27]. Moreover, in young elite endurance runners, the prevalence of amenorrhea has been reported to be even more than 50% and, similarly as in the present study, also associated with lower fat percentage, but not to energy availability from dietary diaries [20,24,25]. Amenorrhea, which is considered to be a long-term marker for low EA, may decrease training adaptations [9], may increase the amount of missed training days [28] and risk of injuries in sports [24], may impair bone health, and may increase the risk of cardiovascular disease [11]. Therefore athletes and their coaches should be better educated about sufficient energy intake and about the risks of trying to maintain too-low body fat percentage for long periods of time. In most of the physique athletes during the off-season, the main goal is to increase muscle size, and this is probably compromised when energy intake is not sufficient [9].

Dietary macronutrient recommendations for physique athletes or bodybuilders in the off-season for protein, fat and carbohydrates are about 1.6–2.2 g/kg, 0.5–1.5 g/kg and ≥3–5 g/kg [2]. The main role of protein is to promote skeletal muscle hypertrophy, fats to support vital body functions, and carbohydrates to provide fuel to support optimal training and glycogen recovery [2,4,5,6]. In our study, the intake of protein of physique competitors was slightly higher in female (3.0 g/kg) and male competitors (2.8 g/kg) that have been suggested to be optimal (1.6–2.2 g/kg) for bodybuilders in the off-season [2] and physique competitors (1.8–2.7 g/kg) [1]. These findings are in line with previous studies where physique athletes have had high protein intake [29]. In a systematic review of competitive bodybuilders, protein intake ranged from 1.9 to 4.3 g/kg for men and from 0.8 to 2.8 g/kg for women [29]. None of the athletes had <1.6. g/kg body weight of protein per day in this cohort, and only one female gym enthusiast had a protein intake of ~1.3 g/kg/day while all the others had at least 1.6 g/kg/day. This suggests that protein intake is more than sufficient in practically all individuals in these sports.

A higher protein intake than recommended does not promote further protein synthesis or gains in lean mass or muscle size [30,31,32], but it also may not have deleterious effects on health in previously healthy young individuals [31,32]. In a study by Antonio et al. [32], resistance-trained participants who consumed a higher amount of protein (4.4 g/kg per day) and energy, compared to a group consuming a lower amount of protein and calories, gained a similar amount of FFM but did not gain additional body fat. Furthermore, in a follow-up study, a group consuming 3.4 g/kg/day of protein gained a similar amount of FFM and lost a greater proportion of body fat compared to a lower protein group, despite a higher energy intake [31]. No negative effects on health were found. Further studies on the effects of higher protein intakes on athletes and gym enthusiasts are warranted to investigate whether better outcomes can be achieved in these populations with more moderate protein intakes.

Previous publications recommended 3–5 g/kg of carbohydrates for bodybuilders [2]. In the present cohort of physique athletes, the average intake was ~4.3 g/kg in males, and in female competitors, ~3.6 g/kg. In female competitors, 30% had a lower-than-recommended carbohydrate intake, while in male competitors, only one had a lower intake. This intake is line with similar studies [29]. A systematic review of the dietary intake of bodybuilders reported large variability in carbohydrate intake with males consuming 243–637 g/day (3–7.2 g/kg), and females 160–415 g/day (2.8–7.5 g/kg) [29]. The intake of dietary fat in female physique athletes (1.0 g/kg/day) and male athletes (0.8 g/kg/day) was in the recommended range (0.5–1.5 g/kg/day) set for bodybuilders [2]. We found that female physique athletes had higher dietary fat intake (g/kg) but a lower intake of carbohydrates compared to male athletes. Additionally, female athletes had a higher intake of protein and lower intake of fat per body weight than gym enthusiasts. The findings were similar, although not statistically significantly different between male athletes and gym enthusiasts. These data indicate that the dietary intake of physique athletes and gym enthusiasts can be considered adequate for physique sports. More studies are warranted to investigate whether low or rather low fat intake in some of the female physique competitors, perhaps through undue emphasis on high protein intake, increases the risk for LEA manifestations such as menstrual disturbances and low BMD.

Our study has a few limitations, including an observational cross-sectional study setting which does not enable the reliable evaluation of causal relationships. Dietary habits were assessed using food recording which is typically prone to ~20% under-reporting in athletes [33], and similarly, reporting exercise and calculating energy expenditures have their limitations. This may affect, e.g., the estimation of EA. In general, physique competitors follow structured diets even during the off-season and compared to other athletes, are more used to estimating portion sizes. Therefore, we consider that the dietary information is more accurate than usual. Additionally, menstrual status was self-reported meaning that females with subclinical menstrual issues (i.e., anovulatory cycles and/or a shortened luteal phase) may have been overlooked in the present study population as pointed out in an earlier study [34]. We also could not control the menstrual cycle phase for testing and thus increasing variation in, e.g., serum estradiol values. Moreover, sleep is a factor that we could not control or measure, which may have affected hormonal pulsatility thus warranting further research in this area. On the other hand, we feel that with our relatively large *n*-size in females, we are now able to report the average values in these athletes and gym enthusiasts throughout the menstrual phase. Unfortunately, we do not have further questionnaires available about the physique training and dieting history of the participants and about the reasons why some of the participants have such a low fat percentage during the off-season. Additionally, the use of supplements by the participants could have affected the measured variables, but our *n*-size does not allow further group analysis as the number of participants not using typical supplements such as protein powders/drinks, creatine, etc., was very small. Finally, our male cohort was small and clearly more LEA studies on males are needed to understand whether males are indeed less sensitive to the reductions in energy availability as some studies suggest [34].

## 5. Conclusions

The dietary intakes, hormonal concentrations, and BMD meet recommendations in most non-dieting female and male physique athletes and are similar to gym enthusiasts in the off-season. Additionally, in female athletes, LEA was rare and did not differ from gym enthusiasts, although amenorrhea tended to be more common in athletes. Furthermore, amenorrhea and LEA were not associated with each other or strongly with any other measured parameters in the present study and, thus, could not be fully explained. However, athletes with amenorrhea and more goal-oriented resistance-training years had a lower fat percentage suggesting that some competitors may artificially maintain unnaturally low body fat percentage for a too-long period of time. Our findings support the view that in the off-season, physique athletes have a dietary intake recommended for the sport, and hormonal and bone health are comparable to non-competitors. Some competitors, however, should perhaps consider having extended off-season periods of higher fat mass and no dieting to restore normal menstrual function.

## Figures and Tables

**Figure 1 nutrients-15-00382-f001:**
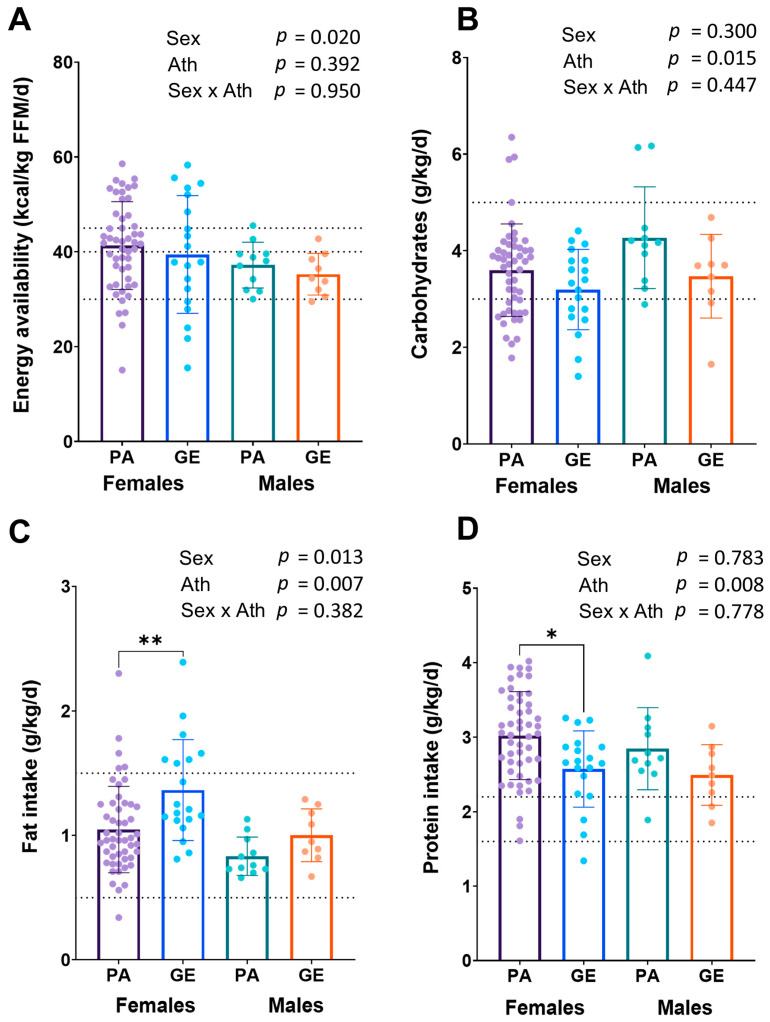
Energy availability (**A**) and macronutrient intakes (**B**–**D**) in physique athletes (PA) and gym enthusiasts (GE). 2 × 2 ANOVA (main and interaction effects) *p*-values are shown as text above the bars and possible post hoc (Tukey’s test) differences in mean values between individual groups. * = *p* < 0.05 and ** = *p* < 0.01. Ath = athlete status (PA or GE). Dashed lines for EA (**A**) show low (<30 kcal/kg FFM) and optimal values (>40/45 kcal/kg FFM) and for macronutrient intakes (**B**–**D**) recommended values for bodybuilders in the off-season [2].

**Figure 2 nutrients-15-00382-f002:**
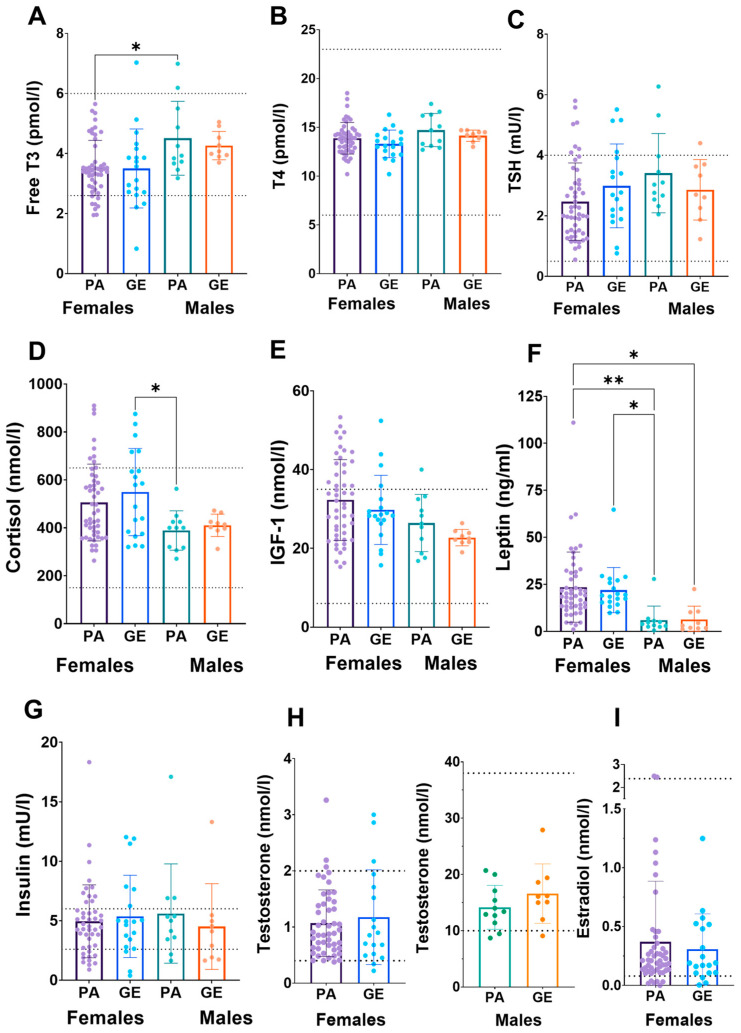
Serum hormone concentrations (**A**–**I**) in physique athletes (PA) and gym enthusiasts (GE). 2 × 2 ANOVA (main and interaction effects) *p*-values are shown as text above the bars and possible post hoc (Tukey’s test) differences in mean values between individual groups. * = *p* < 0.05 and ** = *p* < 0.01. Dashed lines indicate national reference ranges except for leptin, of which reference values are BMI- and sex-dependent and thus not shown. The data were complete (*n* = 89), except for TSH (*n* = 88), insulin (*n* = 85), testosterone (*n* = 87) and estradiol (*n* = 88).

**Figure 3 nutrients-15-00382-f003:**
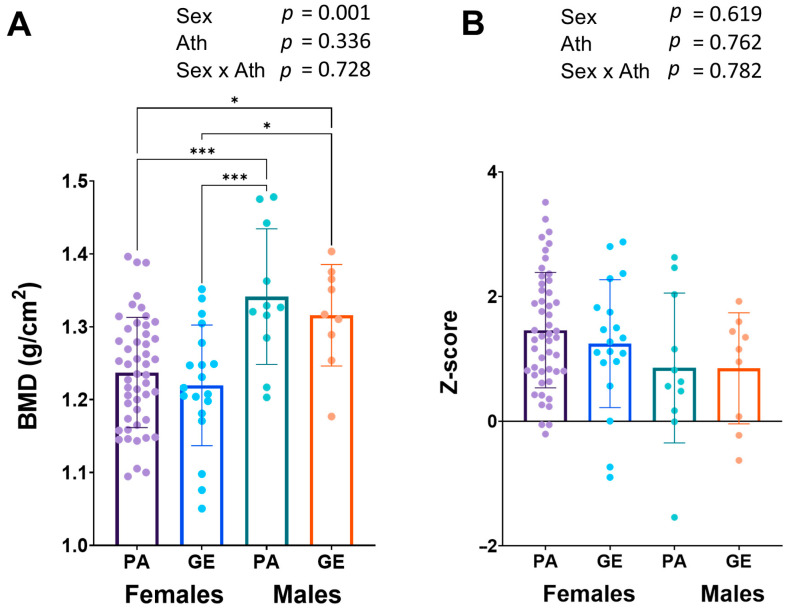
Bone mineral density (BMD) (**A**) and z-scores (**B**). 2 × 2 ANOVA (main and interaction effects) *p*-values are shown as text above the bars and possible post hoc (Tukey’s test) differences in mean values between individual groups. * = *p* < 0.05 and *** = *p* < 0.001. Ath = athlete-status (PA or GE).

**Figure 4 nutrients-15-00382-f004:**
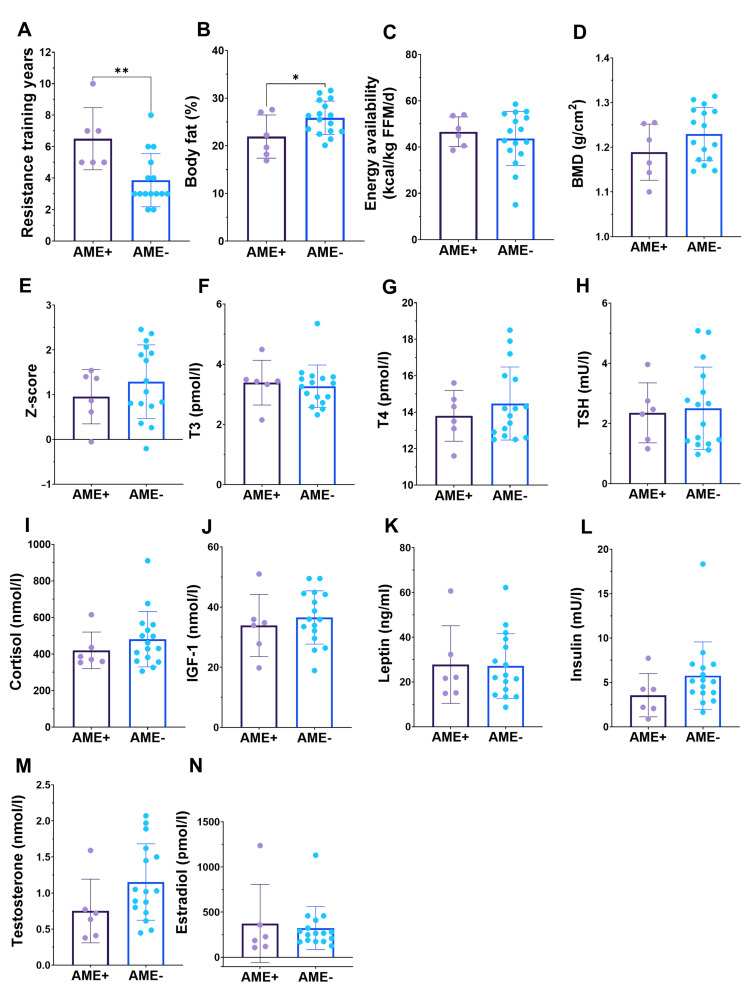
In those athletes who did not use hormonal contraceptives, amenorrheic females (AME+) have similar more resistance training experience in years (**A**) and lower body fat percentage (**B**), but similar energy availability (**C**) bone “health” (**D**,**E**) and hormonal concentrations (**F**–**N**) when compared to those who do not have amenorrhea (AME−). * = *p* < 0.05 and ** = *p* < 0.01.

**Table 1 nutrients-15-00382-t001:** Basic information of the study participants given as mean and standard deviation.

	Females		Males		*p*-Values ^1^		
	PA (*n* = 50)	GE (*n* = 19)	PA (*n* = 11)	GE (*n* = 9)	Sex	Athletes	Sex × Athlete
Age (years)	27.7 ± 4.1	26.4 ± 4.2	28.0 ± 5.6	32.0 ± 4.7	0.085	0.260	0.022
Height (cm)	165.7 ± 5.4	164.7 ± 4.2	180.4 ± 3.9	180.8 ± 2.0	0.001	0.831	0.589
Weight (kg)	65.0 ± 6.9	63.5 ± 5.5	89.1 ± 8.8	86.1 ± 6.0	0.001	0.196	0.685
BMI (kg/m^2^)	23.6 ± 1.7	23.4 ± 1.7	27.4 ± 2.7	26.3 ± 1.7	0.001	0.172	0.429
BF (%)	23.4 ± 5.5	23.5 ± 5.8	14.9 ± 4.2	15.7 ± 6.8	0.001	0.814	0.798
FFM (kg)	50.4 ± 4.7	49.3 ± 4.2	76.7 ± 9.1	72.4 ± 3.8	0.001	0.054	0.255
RT years	4.1 ± 1.9	3.6 ± 1.2	6.5 ± 2.3	6.4 ± 3.7	0.151	0.420	0.546
Training frequency	7.6 ± 3.0	8.0 ± 4.2	5.5 ± 1.6	4.6 ± 1.8	0.688	0.814	0.288
Total training volume (h/week)	6.8 ± 2.5	5.1 ± 2.4	5.0 ± 1.3	4.6 ± 2.2	0.262	0.099	0.688
RT frequency	4.6 ± 0.9	3.5 ± 1.8	4.7 ± 0.8	4.1 ± 1.1	0.194	0.007	0.472
MET (h/week)	41.2 ± 16.9	46.1 ± 35.3	26.6 ± 8.5	29.5 ± 21.7	0.910	0.363	0.710
EEE (kcal)	386 ± 175	418 ± 317	340 ± 114	361 ±261	0.204	0.506	0.796
EI (kJ)	10,230 ± 1820	9846 ± 2102	13,250 ± 1494	12,170 ± 942	0.001	0.124	0.459
EI (kcal)	2444 ± 435	2352 ± 502	3165 ± 357	2907 ± 225	0.001	0.124	0.460
Protein (E%)	32.6 ± 6.8	28.1 ± 5.3	32.1 ± 6.3	29.6 ± 5.0	0.463	0.041	0.591
Protein (g/day)	195.7 ± 40.8	162.9 ± 33.0	251.1 ± 35.8	214.8 ± 38.5	0.001	0.001	0.844
CHO (E%)	38.0 ± 7.2	34.2 ± 5.5	47.7 ± 9.5	40.6 ± 8.1	0.011	0.004	0.422
CHO (g/day)	233.2 ± 67.6	201.2 ± 50.8	376.1 ± 73.3	295.1 ± 61.7	0.001	0.001	0.165
Fat (E%)	25.0 ± 7.1	33.0 ± 6.4	21.1 ± 3.7	26.7 ± 6.0	0.032	0.001	0.466
Fat (g/day)	67.0 ± 19.9	86.0 ± 24.3	73.1 ± 5.8	86.2 ± 19.5	0.516	0.003	0.566

PA = physique athletes; GE = gym enthusiasts; BMI = body mass index; BF = body fat; FFM = fat-free mass, RT = resistance training, E = energy, CHO = carbohydrates. Training refers to all goal-oriented exercise such as resistance training or aerobic training. Training frequency refers to the number of training sessions/week. ^1^
*p* for difference using general linear model.

## Data Availability

The data presented in this study are available on request from the corresponding author after 2023/2024.

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
