# Peer review of "Dietary Intake, Serum Hormone Concentrations, Amenorrhea and Bone Mineral Density of Physique Athletes and Active Gym Enthusiasts"

_nutrients, 2023, doi:10.3390/nu15020382_

Round 1

Reviewer 1 Report

The manuscript is an interesting original research investigating the dietary habits and related health, especially amenorrhea, of the physique athletes and gym trainers during the off-season. Among all contributions to the research field, I would highlight the found results for the amenorrhea condition, which I consider should be further studied.

The manuscript needs some minor improvements prior to publication:

In the abstract, the authors should rephrase the background, which did not put the manuscript in context, include the aim of the study (omitted), and suppress the brackets in line 16.

The authors should include a citation at the end of the statement in Line 33.

The authors should correct the typos in Lines 289, 353, 358.

The authors should consider including a space between the number and the percentage symbol, as indicated in chapter 5 of the International Units Declaration, since Nutrients Instructions for Authors indicate that the SI Units should be used.

Reviewer 2 Report

This study aimed to assess dietary intake, serum hormone concentrations, amenorrhea and bone mineral density of Finnish physique athletes in the off-season and in gym enthusiasts who have similar training goals but have no experience in competing or competition preparation in physique sports. Although this paper was submitted for Nutrients, several concerns are addressed:

-This manuscript is weak and quality of written is poor.

-Abstract: What is aim of study? It is not clear.

-Introduction: No mention regarding to amenorrhea was found. What is importance for athletes?

-Methods: the topic design of study is not correct. 

-DXA did not assess the energy availability as well as Bioimpedance.

-How was assessed the food intake? Database used?

-Results: This is the main problem. Several data has not complete number of subjects assessed. For e.g. hormones.

-Discussion: Few discussed. This is looks like a description of literature not a discussion.

Different of authors, this paper has several limitations. Small sample size, design of study did not allow to achieve the aim proposed, dietary food intake assessed of wrong manner, what was hydration of subjects, use of supplements, hormones?

Round 2

Reviewer 2 Report

This is a very weak study and the aim is not clear. Although the manuscript can be accepted for publication it has limited audience due to poor quality of written.

Author Response

Thank you for your comments, we have made further modifications to the manuscript.